# Mechanical overtone frequency combs

**Matthijs H. J. de Jong** [1,2], **Adarsh Ganesan**[3,4], **Andrea Cupertino**[1], **Simon Gröblacher** [2] **& Richard A. Norte** [1,2] ✉

Mechanical frequency combs are poised to bring the applications and utility of optical frequency combs into the mechanical domain. So far, their main challenge has been strict requirements on drive frequencies and power, which complicate operation. We demonstrate a straightforward mechanism to create a frequency comb consisting of mechanical overtones (integer multiples) of a single eigenfrequency, by monolithically integrating a suspended dielectric membrane with a counter-propagating optical trap. The periodic optical field modulates the dielectrophoretic force on the membrane at the overtones of a membrane's motion. These overtones share a fixed frequency and phase relation, and constitute a mechanical frequency comb. The periodic optical field also creates an optothermal parametric drive that requires no additional power or external frequency reference. This combination of effects results in an easy-to-use mechanical frequency comb platform that requires no precise alignment, no additional feedback or control electronics, and only uses a single, mW continuous wave laser beam. This highlights the overtone frequency comb as the straightforward future for applications in sensing, metrology and quantum acoustics.

Over the last quarter century, optical frequency combs have become key tools for metrology, timing and spectroscopy[1,2], and are indispensable in many laboratories around the world. The fixed frequency and phase relations between the many different tones of a comb have revolutionized fields such as astronomy[3] or cosmology[4], and allowed tests of fundamental physics with atomic clocks[4,5]. Recent developments in optomechanics have expanded methods to create optical frequency combs by interactions with mechanical resonators[6–10]. But in the last few years, a new paradigm of frequency combs has appeared, which are completely mechanical in nature: phononic frequency combs[11–13], also called acoustic or mechanical frequency combs. The ideas behind this field began in nonlinear dynamics, where it was realized that mixing in coupled oscillators may lead to a series of sidebands[14], which can be regarded as a frequency comb if there exists a fixed phase relation[12] between these sidebands. Experimental demonstrations have shown mechanical frequency combs exist in different mechanical systems[10,15–23], and have explored connections to well-known concepts in nonlinear dynamics such as bifurcations[15,18,24], 3- or 4-wave mixing[14,17,25,26], and symmetry-breaking[27]. The fixed

frequency and phase relation between the comb teeth allows the application of techniques known from optical combs. These can, e.g., improve position sensing accuracy in optically opaque materials[28] such as underwater or medical imaging. Mechanical frequency combs can further be used to track and stabilize mechanical resonances[29,30], and enhance Brillouin microscopy[31,32]. Recent proposals from the field of quantum acoustics[33] foresee an important role for devices with multiple mechanical modes with equal frequency spacing. These may also be useful in scaling up transduction between optical and microwave frequencies[34], and engineering the interference between subsequent modes may aid in coupling phonons to multiple qubits[35]. Until now, mechanical frequency combs have been hamstrung by the (generally) nonlinear phononic dispersion relation, which demanded high drive powers, carefully designed mode frequencies or engineered mechanical nonlinearities to obtain an evenly spaced set of modes.

In a different regime of breakthrough physics, optical trapping has allowed us to manipulate and control small particles ranging from single atoms[36,37] to micrometers[38] in size. These particles are confined by the potential created by a strongly focused laser, which has enabled

[1]Department of Precision and Microsystems Engineering, Delft University of Technology, Mekelweg 2, 2628CD Delft, The Netherlands. [2]Kavli Institute of Nanoscience, Department of Quantum Nanoscience, Delft University of Technology, Lorentzweg 1, 2628CJ Delft, The Netherlands. [3]Ahmedabad University, Ahmedabad, Gujarat 380009, India. [4]National Institute of Standards and Technology, Gaithersburg, MD 20899, USA. ✉e-mail: r.a.norte@tudelft.nl

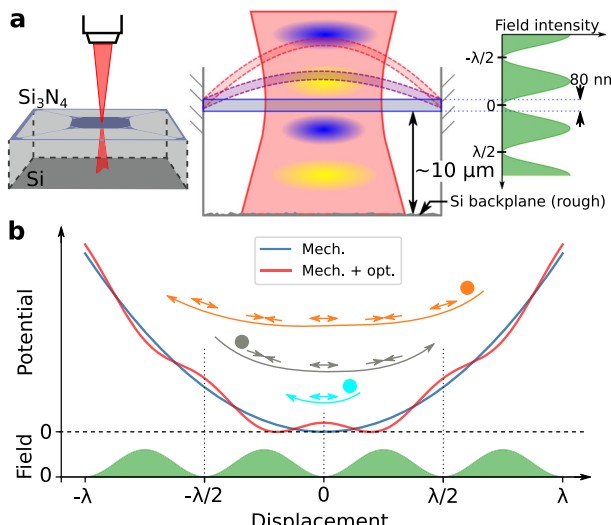

**Fig. 1 | Schematic of overtone frequency comb. a** The suspended Si$_3$N$_4$ membrane is subject to an out-of-plane laser, focused through a microscope objective. Part of the light is reflected from the Si backplane and interferes with the incident light, creating a counterpropagating-wave optical trap such that the field intensity (green) is periodic. The membrane is clamped to the substrate, and motion is predominantly out-of-plane (purple, red dashed lines). **b** The optical field intensity causes a (spatially) periodic modulation of the elastic potential through the dielectrophoretic force (red, blue lines). For the small motion of a membrane at frequency $\omega_0$ (cyan), the restoring force component from the optical field switches sign twice per oscillation (arrow pairs), the same as the elastic potential. However, for larger membrane motion (gray, orange), more optical extrema are crossed so the optical field component switches sign multiple times per oscillation, efficiently generating frequency components at $n\omega_0$ ($n = 2, 3, 4,...$) which form an overtone frequency comb.

the exploration of cutting-edge fields in both fundamental physics and biology. Using optical traps (tweezers), biologists can precisely manipulate anything from single strands of DNA[39] to whole living cells[40]. Optically trapped ultra-cold atoms and levitated nanoparticles are perfect testbeds for fundamental physics involving gravity and mesoscopic quantum mechanics[41–43]. Although optical trapping and frequency combs are widespread techniques, there is little direct overlap between these regimes of physics.

In this work, we uniquely interface optical traps with frequency combs via a mechanism that enables mechanical frequency combs without requiring feedback control, external drives and frequency references, or precision optics. We observe that due to a weak counterpropagating optical trap[44,45], strongly driven silicon nitride (Si$_3$N$_4$) membranes vibrate not only at their mechanical eigenfrequencies, but at perfect integer multiples of a single frequency, which can form a mechanical frequency comb. The standing-wave optical field exerts a dielectrophoretic force on the membrane, which is modulated by the membrane's motion as it crosses extrema of the optical field. If the displacement is small, this interaction can suppress mechanical dissipation[46], but if the displacement becomes of the order of a quarter wavelength, it creates integer multiple copies of the original membrane motion ("overtones"[47], see Supplementary section A). These overtones share a fixed frequency and phase relation, and thus form a frequency comb while avoiding all difficulties of engineering a linear mechanical dispersion relation. Strikingly, the overtones are solely dependent on the amplitude of motion and the optical field, and thus independent of the drive mechanism. We utilize an optothermal parametric drive based on the same single, unmodulated optical field to bring the membrane to self-oscillation. This allows us to build a mechanical comb without any additional pump tone or frequency

reference, which makes overtone combs uniquely simple to generate. We will first describe the mechanism that creates the overtones and study their behavior in the frequency domain. Then, in the time domain, we will show the fixed phase relation that makes the overtones act as a frequency comb.

## Results

### Overtone and driving mechanism

In this section, we first describe the physical system and the mechanism that creates the overtone frequency combs, and then introduce a more quantitative model for their dynamics. The system consists of a suspended $t = 80$ nm thick Si$_3$N$_4$ trampoline membrane[48] (see Methods), shown schematically in Fig. 1a. It rests ~10 μm above a backplane formed by the silicon (Si) substrate, and a laser ($\lambda = 633$ nm, $p \lesssim 3$ mW) from a commercial Polytec MSA400 laser Doppler vibrometer (LDV) is incident on the membrane. At this wavelength, the Si$_3$N$_4$ reflects $\lesssim$30% of the light, and Si reflects about 35%. Part of the light thus forms a standing wave, Fig. 1a, periodic in the direction of the mechanical motion.

The dielectric Si$_3$N$_4$ experiences a dielectrophoretic force proportional to the gradient of the optical intensity, similar to a particle in a counterpropagating-wave optical trap[44,45]. The trap also exerts a radiation pressure force, but this is negligible in our system (ref. [6] and Supplementary section C). If the dielectric moves (e.g., by driving a mechanical eigenmode), it will experience a restoring force from its own elastic potential (blue solid line in Fig. 1b), with an additional component from the optical field (red solid line). For small motion ($|x| \ll \lambda/4$, cyan line in Fig. 1b), the optical potential functions as an additional spring[46]. However, if the motion of the membrane is of the order of the optical potential period ($\lambda/2$), the modulated potential generates the overtones.

This can be seen as follows: the restoring force of the mechanical potential switches sign twice per oscillation (arrow pairs in Fig. 1b). If the motion is large enough (gray, orange lines) for the resonator to cross multiple extrema of the optical field, the optical component of the restoring force switches direction $2n$ times per oscillation ($n$ integer number of optical extrema). If the original motion was at mechanical eigenfrequency $\omega_0$, this effect generates motional components at $n\omega_0$, which are the overtones of the original eigenmode. We will show in the section "Comb dynamics" that these overtones have a fixed phase relation and thus form a mechanical frequency comb. Because the overtones originate from the combination of the optical and mechanical potential, they completely avoid the difficulties of engineering the mechanical dispersion while still resulting in perfectly evenly spaced tones. In contrast to other combs, the tones do not exist around some carrier frequency.

We model the overtone frequency comb using a resonator described by displacement $x(t)$ ($x = 0$ mechanical equilibrium), with resonance frequency $\omega_0$ and decay rate $\gamma$, where we have divided by the simulated effective mass $m_{\text{eff}} \simeq 12 \times 10^{-12}$ kg. We write the dielectrophoretic force as a proportionality constant $F_o$ (units of force) times the gradient of the periodic part of the optical intensity, $\nabla E^2 \propto \sin\left(\frac{4\pi}{\lambda}(x - x_{\text{off}})\right)$. This way we can move finite-size effects of the membrane into $F_o$, which we numerically evaluate in Supplementary section C. We obtain the equation of motion

$$\ddot{x} + \gamma\dot{x} + \omega_0^2 x = \frac{F_o}{m_{\text{eff}}} \sin\left(\frac{4\pi}{\lambda}(x - x_{\text{off}})\right), \quad (1)$$

which requires only a suitable initial condition $|x|_{t=0} \gtrsim \lambda/4$ to demonstrate the creation of overtones. This condition is much larger than typical interferometric position measurements, which is why we use a LDV that is capable of resolving such large displacements (see Methods). Equation (1) also shows that the overtones are independent of the choice of drive (e.g., piezoelectric, electrostatic, thermal). By utilizing

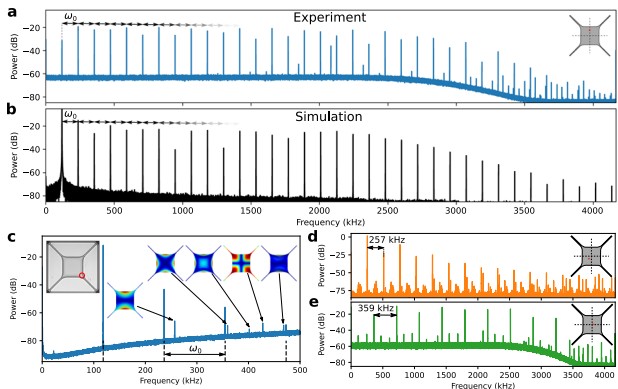

**Fig. 2 | Overtone frequency combs. a** Measured spectrum of a mechanical frequency comb of the fundamental mode ($\omega_0 = 2\pi \times 118.049\,\text{kHz}$) of a suspended Si$_3$N$_4$ membrane, consisting of 35 peaks spaced by $\omega_0$. The overtones are spectrally flat until ~2200 kHz, after which their amplitude drops exponentially. Inset shows the location of the laser spot to generate and read out the comb. The additional peak around 2083 kHz is from a different mechanical mode, see Supplementary section F. **b** Simulated overtone comb spectrum, by integrating Eq. (1), matching the measurement of **a. c** Simultaneous measurement of fundamental-mode frequency comb (first four teeth) and higher-order mechanical modes of the membrane. Insets: laser spot position (left) and mode shapes. **d, e** Mechanical frequency combs of the second mode (257 kHz) and third modes (359 kHz). Difference in the noise floor is due to different decoders used due to overloading. The 257 kHz mode has a near-degeneracy, so the comb spectrum is less clean. Insets show the laser position to drive and read out the different modes.

the spatially-periodic optical field through optothermal effects[49], we can create a parametric drive powerful enough to bring the membrane to self-oscillation. That is, when the resonator moves through the field, the optical intensity it experiences is modulated at twice the frequency of the original motion. This modulates the resonator frequency through absorption (thermal expansion), thus creating an optothermal parametric drive (see Supplementary section D). Our membranes are patterned with a specific photonic crystal that enhances the absorption of 633-nm laser light. This allows a single, continuous-wave laser beam of mW power to bring the membrane into self-oscillation. While we verify that the overtones can also be generated via inertial (piezo) driving (see Supplementary section E), we leverage the optothermal drive to avoid any external pump tone or frequency reference. This property of overtone combs is unique within the mechanical frequency combs, and allows for significantly simpler setups.

## Overtone frequency comb
We measure the overtone frequency comb for the fundamental mode $\omega_0 = 2\pi \times 118.049\,\text{kHz}$ of our membrane in Fig. 2a. In the velocity power spectrum, we see a series of 35 peaks spanning the full detection bandwidth of the setup (4.2 MHz), spaced by $\omega_0$. This spacing is perfectly uniform, limited by the spectral resolution of the setup to $4.7 \times 10^{-8}$ relative spacing difference. A comparison of this comb to other mechanical combs is included in the Supplementary section A. Through the integration of the velocity signal, we verify the displacement $x \gg \lambda/4$. By numerically integrating Eq. (1), we can reproduce this overtone comb, shown in Fig. 2b, using only the optical potential strength $F_o$, offset $x_{off}$ and initial conditions [$x_0, v_0$] as fit parameters (assuming steady state, so $\gamma = F_d = 0$). This highlights that the nonlinearity comes from the optical field, without introducing mechanical nonlinearities previously used to explain this behavior[47].

The overtones can be distinguished from the other mechanical eigenmodes of the membrane. The mechanical eigenmodes can be detected and identified (Fig. 2c). When the overtone comb is generated, additional peaks appear in the spectrum (dotted black lines). This demonstrates that the overtones are not affected by the mechanical

dispersion relation and do not require engineered nonlinear resonances[12,17]. Furthermore, it is possible to generate combs from the second and third mechanical eigenmodes, by selecting the right laser position on the membrane, Fig. 2d, e. This makes the frequency spacing variable, limited by our ability to drive a particular eigenmode. Finally, for some laser positions the overtone comb at $\omega_0$ interacts with a different mechanical eigenmode ($\omega_h$), which could allow an extension of the comb bandwidth (Supplementary section F).

We examine Eq. (1) to better understand the behavior of this mechanism (details in Supplementary section A). Firstly, when the displacement $x$ increases, more overtones appear due to the increasing number of optical extrema crossed. Each overtone will again be modulated, so they have equal power up to a certain cutoff, which is beneficial for many applications (~2200 kHz in Fig. 2a, b). Secondly, the strength of the optical field ($F_o$) controls the power of each overtone relative to the original mode $\omega_0$. Finally, the offset $x_{off}$ between mechanical and optical zeros determines the relative intensity of the odd and even number overtones. The position offset $x_{off} \simeq 40\,\text{nm}$ is consistent between different membranes in this work, and is half the thickness of the Si$_3$N$_4$ membranes. We can repeatedly create the overtone comb in different membranes, and study the effect of the optical beam itself on the membranes and overtone comb in Supplementary section G.

## Comb dynamics
The comb dynamics can be visualized by starting a measurement with the laser spot positioned away from the membrane, as shown in Fig. 3a. We move the laser to the membrane center at $t = 8.7\,\text{s}$ such that the optothermal parametric driving starts increasing displacement. We analyze the dynamics by cutting the recorded time signal into intervals and performing a Fourier transform on each. Then we concatenate the spectra such that we can study their behavior over time (Fig. 3b) and monitor the power of individual overtones by taking linecuts (Fig. 3c).

In Fig. 3b, the fundamental mode shows a 9 Hz upwards shift in frequency, which is reproduced as an $n \times 9\,\text{Hz}$ upwards shift for the $n$th overtone (e.g., $n = 5, 15$ in the figure) and likely originates from slow thermalization (Supplementary section D). This illustrates thermal tuning would be an effective mechanism to control and tune the comb spacing, without compromising comb uniformity.

We plot the power of the overtones as the comb grows in Fig. 3c. We can simulate and reproduce this growth quantitatively by integrating Eq. (1). Comparison between the experimental data (markers) and simulation (solid lines) in Fig. 3c shows good agreement with fit parameters [$F_o = 3.8\,\text{pN}$, $x_{off} = 40\,\text{nm}$, $x_0 = 5\,\text{nm}$, $v_0 = 1\,\text{nm s}^{-1}$, $\gamma/2\pi = 0.8\,\text{Hz}$ and $F_d = 2.1\,\text{pN}$]. Most important for the overtones is $F_o$, which is in excellent agreement with simulated force on the order of pN (Supplementary section C). All extracted curves share a vertical offset to account for the total detection efficiency. This shows Eq. (1) reproduces the dynamics of the resonator and individual overtones.

In Fig. 3d, e, we observe the membrane motion from the thermal state at $t < 5\,\text{s}$ to the steady state of the comb at $t > 33\,\text{s}$. Initially, we detect only the fundamental mode $\omega_0$, until the resonator starts crossing multiple optical extrema. Higher overtones appear as the maximum displacement grows. In contrast to Fig. 2a, the fundamental mode is the most powerful. The detection efficiency likely varies for each overtone due to its shape[47]. The displacement stops growing at $t \simeq 26\,\text{s}$, and decreases slightly before the system reaches a steady state (see Supplementary section H). It is likely limited in amplitude by a mechanical non-linearity, but the amplitude overshoot and the oscillation of amplitude of overtones suggest that the interaction with the optical field plays a role. In the steady state, the fractional frequency stability of the 30th overtone is $7.5 \cdot 10^{-10}$ over a 6-h period (Supplementary section H), limited by thermal drifts. The frequency of the overtones is determined solely by the mechanical frequency, and is thus not affected

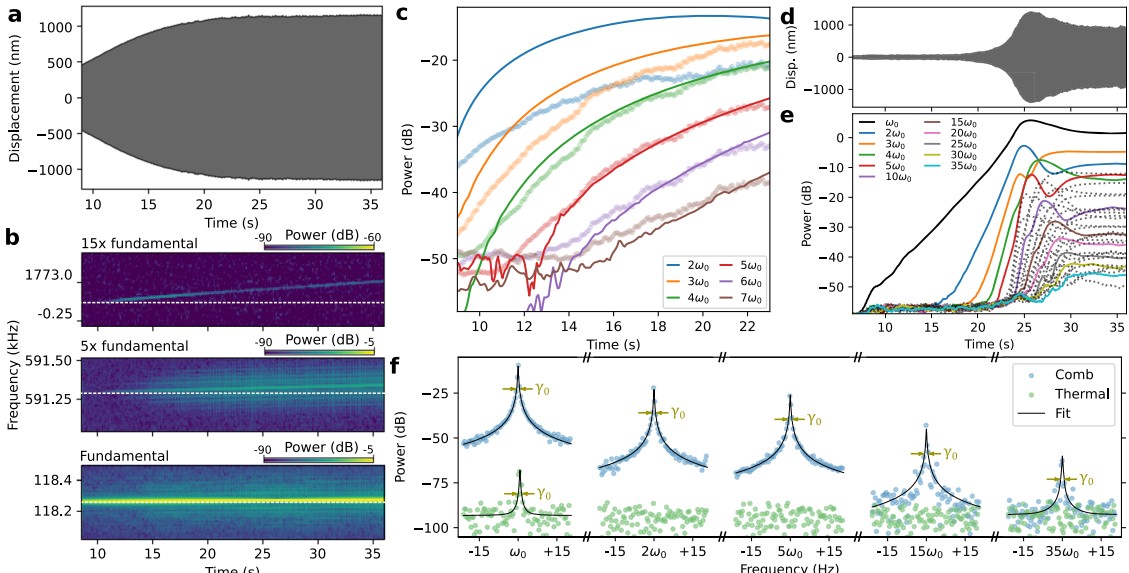

**Fig. 3 | Comb dynamics. a** Measured displacement of membrane, showing clear growth before reaching a plateau around $t = 24$ s. **b** Measured spectrum of membrane motion close to fundamental mode and two of its overtones (5th and 15th). A thermal shift of 9 Hz of the fundamental mode is visible as a $5 \times 9$ and $15 \times 9$ Hz shift in the overtones (white dashed lines are horizontal to guide the eye). **c**: First six overtones increase in power as the membrane displacement increases (markers) in **a**, with simulated dynamics (solid lines) based on the integration of Eq. (1) matching quantitatively to the highest four. **d** Membrane displacement of a different device (different chip) than **a**, showing growth from close to the thermal regime to steady state. **e** Overtone amplitudes extracted from the spectrum of measurement of **d**, showing all 35 overtones within the detection bandwidth (black, dotted lines are overtones with numbers between the labeled ones). **f** Spectrum showing measured fundamental mode and some selected overtones in the comb regime (blue) and thermal (green), along with Lorentzian fit with center frequencies $n\omega_0$ ($n$ integer) and identical linewidths $\gamma_0$.

by drifts in the laser frequency. Changes in the laser power will affect the overtone amplitudes, mainly via the optothermal parametric driving. The uniformity of the comb is not affected by mechanical frequency shifts (nor by the laser), thus uniformity is constant.

We isolate and plot the spectrum around several of the overtones in Fig. 3f. In the thermal regime (green markers), only the fundamental mode is visible and we can fit a Lorentzian with linewidth $\gamma_0 = 2\pi \times 0.07$ Hz to the peak. For the comb in a steady state, the fundamental mode retains its linewidth $\gamma_0$, and we can derive that the comb mechanism does not cause additional noise, Supplementary section I. All overtones possess the same linewidth $\gamma_0$, which does not match the $n\gamma_0$ scaling expected from a frequency-fluctuation limited linewidth, but matches a decay-rate limited system. These properties combined should allow frequency combs with single-mHz linewidths based on ultra-high-Q membrane resonators[50].

To finally show that the overtones form a comb, i.e., that a fixed phase and frequency relation exists, we show the time-domain signal in Fig. 4a. Unlike combs centered around a carrier (such as soliton-based mechanical[17] and optomechanical[8] frequency combs), there is no component with periodicity longer than the mechanical period $2\pi/\omega_0$ (Fig. 4b, c). This highlights the different physical and dynamical processes behind the overtone comb. If the fundamental mode is the dominant component in the comb, we get a sinusoid (Fig. 4b, c, blue line). When other components are dominant, particularly $2\omega_0$ and $3\omega_0$, we get the peaked curves (Fig. 4c, green and orange lines), which retain the periodicity of the fundamental mode. The shape of these curves proves that the entire overtone comb is phase-coherent, as they are formed by a sum of cosines with the same phase offset (see Supplementary section J). This behavior is retained not only in the steady state, but also during comb growth, Fig. 4d. There we plot the time signal at various stages during the measurement of Fig. 3d, e, which shows a smooth transition from sinusoidal to peaked behavior as the comb grows. Thus the overtones form a mechanical frequency comb.

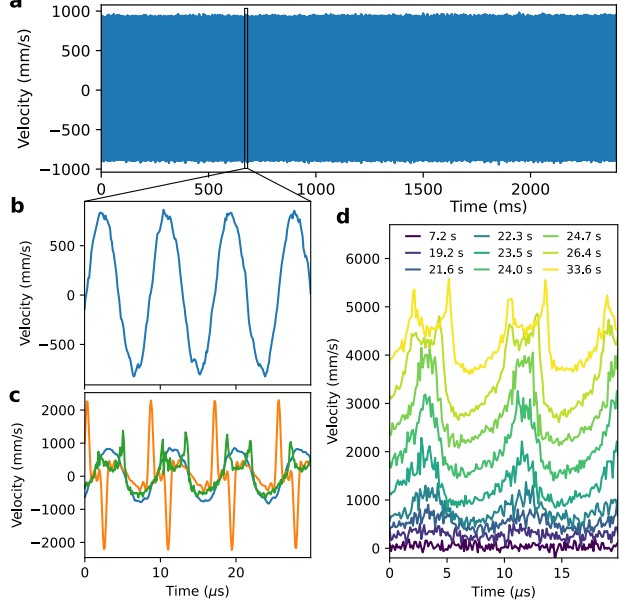

**Fig. 4 | Time-domain frequency comb. a** Measured velocity of resonator in frequency comb regime, displaying no pattern with a period longer than $1/\omega_0$. **b** Zoom-in of time-domain signal in **a**, showing the 118 kHz fundamental mode dominates. **c** Three different measured time signals in steady state, with varying relative strengths of the overtones (blue: $\omega_0$ dominates, orange: $2\omega_0$ and $3\omega_0$ dominate, green: $1\omega_0$ to $6\omega_0$ similar in strength). All traces can be reproduced only with phase-coherent addition of the different overtones. **d** Time-domain signal of comb during growth, the same measurement as Fig. 3e at the indicated times. This shows the smooth transition from the thermal regime (bottom) to the comb in a steady state (top), traces are offset vertically for clarity.

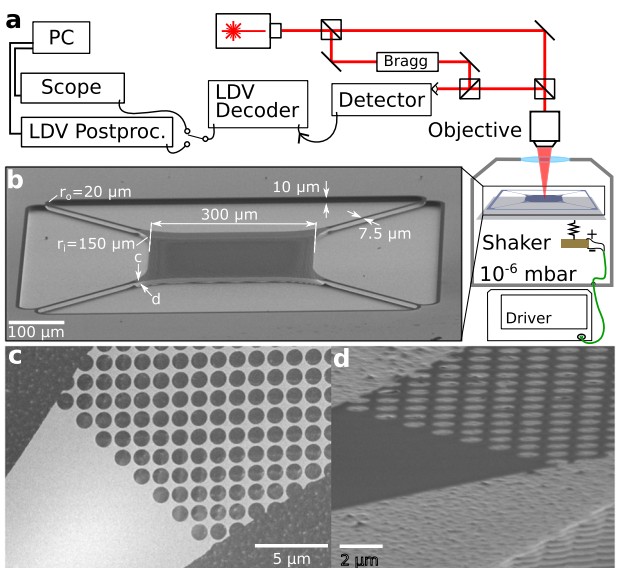

**Fig. 5 | Setup and membranes. a** Schematic of the laser Doppler vibrometer (LDV) setup, where we access the chip containing the membranes using a microscope objective directly outside a vacuum chamber. **b** SEM image of the suspended membrane, with nominal design parameters in white. **c** Top-down view of a corner of the membrane, highlighting the interface of the photonic crystal with the tether. **d** Tilted view to show Si surface below **c**, to illustrate etch roughness and an etched imprint of the photonic crystal pattern (bottom right).

## Discussion

We have discovered a mechanism that uniquely interfaces two break-through concepts: optical trapping and frequency combs. This allows for mechanical frequency combs whose simplicity stands out from those based on previous mechanisms. This is realized by integrating a suspended dielectric membrane in a weak optical trap. The dielectrophoretic force from the optical field modulates the mechanical potential of the membrane. This modulation creates integer multiple copies (overtones) of the membrane's motion, which forms a frequency comb. We show combs of up to 35 overtones in a 4.2 MHz bandwidth with control over the frequency spacing, excellent uniformity, stability and no added mechanical noise. The integration of the membrane in the optical trap allows us to combine the overtone comb with optothermal parametric driving, which brings the membrane to self-oscillate. We thus realize a frequency comb that requires no external drive or control frequencies and which uses a minimal setup (laser, microscope objective and vacuum chamber). This makes it more versatile and easier to use than other ways of generating these combs. In summary, this mechanism unlocks the potential of mechanical frequency combs for sensing, timing and metrology applications at the microscale, and provides native integration with phononic circuits.

## Methods

### Laser Doppler vibrometer

The setup used in this work is shown in Fig. 5a. It consists of a commercial LDV, Polytec MSA400, which is depicted schematically. Light from the LDV goes through a microscope objective, which focuses it on the chip containing the membrane resonators. This chip is placed in a vacuum chamber and pumped down until the pressure is $<5 \times 10^{-6}$ mbar, to reduce gas damping. There is a piezoelectric shaker mounted to the sample holder, by which we can drive the membrane, though for the majority of the measurements we use the thermal parametric driving described in the Supplementary section D. The reflected light from the membranes is Doppler-shifted due to their out-of-plane motion, which is then detected by the LDV. To extend the time we can

continuously measure, we add a Rohde & Schwarz RTB2004 digital oscilloscope to read out the LDV decoder. Using the history function of this oscilloscope, we can chain 16 measurements of 20 million data points each, which allows 38 s of time signal at 8.33 MHz sampling rate. However, this method for reading out the velocity comes at a cost of the calibrated readout that the LDV postprocessing offers.

The LDV outputs a voltage signal proportional to velocity or displacement, depending on the LDV decoder used. If using the LDV postprocessing, it can easily be Fourier-transformed and exported. If using the oscilloscope and history function, we sequentially read out the measurements afterward and concatenate them in the correct order in postprocessing. We can then further extract information from this signal either by integrating (to obtain the displacement), or by using Scipy's short-time Fourier transform function to obtain the time-varying behavior of the comb spectrum.

### Trampoline membranes

The membrane structures used in this work are fabricated out of 100 nm thick stoichiometric low-pressure chemical vapor deposition $Si_3N_4$ on top of a 1-mm Si chip. This was done by writing the pattern using electron beam lithography and an inductively coupled plasma (ICP) etch to transfer that pattern to the $Si_3N_4$. The membranes are then released using a second ICP etch, at $-120\,°C$ for 30 s, resulting in about 10 μm of undercut and a final $Si_3N_4$ thickness of 80 nm. An SEM image of a released trampoline is shown in Fig. 5b, with the nominal design parameters added in white. The resulting trampolines have well-characterized mechanical properties[48,51], with fundamental mode frequencies at 120 kHz and Q-factors typically of 1 million.

The membranes are pattered with a periodic array of holes (Fig. 5c) that forms a photonic crystal, though it is not designed to have a high reflectivity at the operating wavelength of our LDV ($\lambda = 633$ nm). Instead, the holes function as etch release holes to evenly release the membrane in the final ICP etch, which is essential for the fabrication yield. For these membranes, the periodic array of holes functions as a method to control the mass and resonance frequencies[51], and possesses internal optical resonances that facilitate absorption which we detail in Supplementary section D. Finally, the release etch imprints the photonic crystal pattern on the Si backplane, which roughens the surface. Fig. 5d shows the final Si surface roughness from the release etch (top left half) and the imprint below the periodic hole pattern (bottom right half).

## Data availability

The data (raw data, analysis and calculation scripts, and finite element simulations) that support the findings of this study are available at https://doi.org/10.4121/19821016.v1.

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

## Acknowledgements

We would like to acknowledge Farbod Alijani, Peter Steeneken, Lior Michaeli, and Albert Schliesser for interesting discussions, Dongil Shin for help with the thermal simulations, and Wouter Westerveld and Gerard Verbiest for their comments on the manuscript. R.A.N. would like to acknowledge support from the Limitless Space Institute's I2 Grant. M.H.J.d.J., A.C., S.G., and R.A.N. acknowledge valuable support from the Kavli Nanolab Delft and from the Technical Support Staff at PME, 3mE Delft, in particular from Gideon Emmaneel and Patrick van Holst.

## Author contributions

M.H.J.d.J. performed the experiments, simulations and data analysis. A.G. and M.H.J.d.J. developed the theory. A.C. developed the fabrication process and fabricated the devices. M.H.J.d.J., A.G., S.G., and R.A.N. developed the interpretation of the results. R.A.N. supervised the project. All authors contributed to writing and editing the manuscript.

## Competing interests

The authors declare no competing interests.
