## [Peer review file · Nature Communications]

REVIEWER COMMENTS

Reviewer #1 (Remarks to the Author):

In this manuscript, the authors have demonstrated mechanical overtone frequency combs, by monolithically integrating a suspended dielectric membrane with a counter-propagating optical trap generated via its own substrate. The standing-wave optical field exerts a dielectrophoretic force on the silicon nitride (Si₃N₄) membrane, which is modulated by the membrane's motion as it crosses the extrema of the optical field. The periodic optical field modulates the dielectrophoretic force on the membrane at integer multiples of the membrane's frequency of motion. As a result, strongly driven silicon nitride SiN membranes vibrate not only at their mechanical eigenfrequencies but at perfect integer multiples of a single frequency, thus forming a mechanical frequency comb.

Phononic Frequency Combs have been proposed through Nonlinear Resonances [PRL 112, 075505 (2014)] and realized via, e.g., intrinsic three-wave mixing [PRL 118, 033903 (2017)]. In addition to the intrinsic mechanical nonlinearity, by controlling dielectric gradient forces, the resonance frequencies of the flexural in-plane and out-of-plane oscillation of the high-stress silicon nitride string could also be tuned, introducing the nondegenerate parametric driving. The dielectric gradient force acting on the SiN membrane has been demonstrated by employing two adjacent gold electrodes [PRL 118, 254301 (2017)], or using the piezoelectric ring actuator [PRL 122, 154301 (2019)]. In this work, the authors present another method, the dielectrophoretic force from the optical field, that enables a mechanical frequency comb without requiring feedback control, a frequency reference, or precision optics.

In conclusion, I think the experiment results are presented clearly. I believe that the authors did a solid experiment and performed a reasonable analysis. However, on the physics side, the experimental results are not very surprising. The self-sustained oscillator and frequency comb through dielectrophoretic force have been extensively studied and used to study frequency combs. In terms of application, I don't think this work has a significant impact on micro combs. Several vital factors of micro combs are not mentioned in the manuscript, e.g., size, bandwidth, resolution, and self-referencing. Hence, I recommend that revisions should be made to the manuscript.

Other minor things:

- (1) As shown in figure 1, the simultaneous generation of multiple sets of frequency modes should be avoided during typical frequency comb systems designing. I hope the authors could explain these modes around the 4000 kHz, and how to avoid them.
- (2) Fig1 d shows quite a lot of spurious modes, of which the peak heights are comparable and even higher than frequency combs (after the 7th sidebands). This will affect the real case application.

(3) I see this work use a trampoline structure SiN resonator. What is the advantage compared to the case that directly uses a whole membrane? Presenting a table of the parameters of the device (e.g., mechanical mode frequency, linewidth, Q value) may improve the readability of the article.

(4) Can the authors clarify whether the trapped optical field will introduce interactions between different sets of the frequency combs, e.g., interactions between the mechanical frequency combs of the second mode (257 kHz) and third modes (359 kHz).

Reviewer #2 (Remarks to the Author):

This paper reports the high overtone generation in a SiN mechanical resonator. The authors use a periodic optical potential generated by the standing wave formed between the substrate and resonator surfaces to modulate its motion and very high-harmonic overtones were observed when the displacement of resonator becomes larger than the wavelength. They performed interesting experiments and the manuscript is well described with reliable theoretical analysis. However, I meet the following essential issues and do not recommend the publication at least with the present form.

1. They use the wording "frequency comb" to describe the observed multiple and equally-spaced spectrum but I am not sure if this terminology is adequate. It is, of course, a matter of definition but majority of frequency comb experiments consist of a high frequency carrier with multiple sidebands with low-frequency interval, which is highly different from the carrier frequency. In contrast, the generated multiple peaks in this work are all high harmonic overtones and I wonder if we can refer them as a frequency comb. It is rather natural to use the words something like "ultrahigh overtone generation".

2. Relating to this first point, the application of this system is unclear. In the introduction part, the authors list up the applications of optical frequency comb, but these applications utilize equally spaced sidebands, not the high harmonic overtones of carrier signals. One of the significant features of frequency comb is the generation of pulse train (or "solitons"), but I believe that the overtones only modify the shape of the waveform from sinusoidal to distorted one, as shown in Fig.4c and d, and do not generate a longer-period pulse train. It is also the case in spectroscopy usages (Ref. [11] and [12]), where I cannot imagine the mechanical counterpart; how to utilize the overtones in the "single" mechanical resonance in, for example, molecular sensing etc. ? The use of this overtone generation should be something different from that of optical frequency comb. I recommend the authors to modify the introduction and the conclusion parts to justify the importance of "overtone generation".

3. The role of stability of high overtones should be discussed. The stability of frequency comb is managed by system non-linearity or feedback mechanics so that it is worth being characterized and

analyzed. However, the stability of this overtone "comb" simply reflects that of fundamental frequency generated by the parametric driving. In other words, the overtone generation itself does not contribute to stabilize the frequency accuracy. If the stability of the used laser is low, the harmonics have worse stability but cannot be better than the carrier oscillation. It is unclear for me how to use the same or worse stability of the mechanical overtones in real timing device applications.

4. The authors use a Doppler interferometer for detecting the acoustic vibration. It works when the displacement is larger than the wavelength but it is also true that apparent high harmonic overtones are introduced in the output signal with large displacement. Used electronics also can sustain the overtones and the author should experimentally prove the contributions are enough small.

More details to be answered:

5. It is interesting to see the lateral mode shape of the high overtones. Because it is not the eigenfrequency of the resonator, the lateral shape might be different from that of fundamental frequency (see for example ref. [48]). This can be a new aspect and I recommend the authors to discuss it as well as the validity of eq (1).

6. On the piezo drive shown in the S.I., the white noise excitation might directly excite the overtone oscillation. It is necessary to experimentally confirm that similar intensities of overtones can be generated when only the fundamental mode is piezo-excited.

7. It is also interesting if the optical parametric drive excites only the fundamental frequency. If I correctly understand the drive mechanism, the parametric drive is not the blue side pumping but the laser light creating the thermal stress with a kind of positive feedback to induce a degenerate parametric drive. Therefore, there is no externally given frequency and any higher mode can be excited.

8. In the spectrum Fig.2 a, the amplitude of fundamental frequency is -10dB lower than the overtones. I do not understand how this situation is created. Does it related to the points mentioned above?

In summary, although they performed nice experiments and analysis, it is quite unclear for me the importance of this work in real applications. The comparison with optical frequency comb is not suitable to my opinion because the intrinsic properties are different, so that the importance of high overtone generation should be justified. With the present form of manuscript, the importance seems to be not in its application as frequency comb but in the new aspects as nonlinear systems, where very high overtones might create new dynamics of mechanical oscillation but the analysis is not sufficient for justify the publication.

We would like to express our gratitude to the reviewers for their comments and consideration. We have copied the reviewers' remarks below, and addressed them. There have been significant changes made to the manuscript (marked in blue), as well as additional measurements further supporting our claims. Some of these measurements were done based on suggestions of Reviewer 2, for which we like to thank them. We feel we have addressed the major issues regarding novelty of the work raised by Reviewer 1, and have answered the technical questions/remarks. The conceptual issue of whether the overtones form a frequency comb (raised by Reviewer 2) has also been addressed, both in our reply and in the revised manuscript. In particular the introduction and conclusion have been rewritten to focus on the applications of mechanical frequency combs, rather than their optical counterparts.

Reviewer 1:

Phononic Frequency Combs have been proposed through Nonlinear Resonances [PRL 112, 075505 (2014)] and realized via, e.g., intrinsic three-wave mixing [PRL 118, 033903 (2017)]. In addition to the intrinsic mechanical nonlinearity, by controlling dielectric gradient forces, the resonance frequencies of the flexural in-plane and out-of-plane oscillation of the high-stress silicon nitride string could also be tuned, introducing the nondegenerate parametric driving. The dielectric gradient force acting on the SiN membrane has been demonstrated by employing two adjacent gold electrodes [PRL 118, 254301 (2017)], or using the piezoelectric ring actuator [PRL 122, 154301 (2019)]. In this work, the authors present another method, the dielectrophoretic force from the optical field, that enables a mechanical frequency comb without requiring feedback control, a frequency reference, or precision optics.

In conclusion, I think the experiment results are presented clearly. I believe that the authors did a solid experiment and performed a reasonable analysis. However, on the physics side, the experimental results are not very surprising. The self-sustained oscillator and frequency comb through dielectrophoretic force have been extensively studied and used to study frequency combs.

We would like to thank the reviewer for their comments on the presentation, experiments and analysis. We agree completely that the existence of the dielectrophoretic force (gradient force) is not new, but we are not aware of any literature showing it can lead to a mechanical frequency comb. In PRL 118, 254301 (2017), the dielectrophoretic force couples two mechanical modes, but the appearance of the frequency comb requires a parametric drive rather than the resonator moving through a static potential, as in our work. In PRL 122, 154301 (2019), the resonator is driven inertially (by a piezo shaker). Although their resonator and setup could conceivably lead to the same mechanism as we describe, the authors of PRL 122, 154301 (2019) propose an explanation involving multiple non-linear mechanical modes.

Our claim is not the newness of the dielectrophoretic force, but the fact that it leads to a mechanical frequency comb through a spatial modulation of the mechanical potential. Additionally, the optothermal parametric drive (using the same laser) allows us to avoid any external drive, which is unique amongst the mechanical frequency combs. This optothermal parametric drive has been enabled by careful engineering of a photonic metamaterial on our trampolines. The ability to use the same laser to both provide a non-linear potential and a drive mechanism simultaneously set our platform apart; making them much easier to operate and miniaturize.

We have significantly re-written the introduction of the work to highlight the difference between our work and the current available literature on mechanical frequency combs, and have reworded our explanation of the effect to more precisely delineate the differences with other mechanisms.

In terms of application, I don't think this work has a significant impact on micro combs. Several vital factors of micro combs are not mentioned in the manuscript, e.g., size, bandwidth, resolution, and self-referencing. Hence, I recommend that revisions should be made to the manuscript.

The main benefit of the overtone comb over other mechanical frequency combs is that it completely removes the need for an external drive. This greatly simplifies the operation and setup required for such a frequency comb. Additionally, since the overtone comb does not have nor require a carrier signal, self-referencing should be trivial: if f is the fundamental mode frequency, $2f$ will simply be the first overtone.

We have amended the manuscript to include a table of relevant mechanical frequency comb properties (Table S1), and have changed the wording in the main text to emphasize the benefits of the overtone comb.

Other minor things:

(1) As shown in figure 1, the simultaneous generation of multiple sets of frequency modes should be avoided during typical frequency comb systems designing. I hope the authors could explain these modes around the 4000 kHz, and how to avoid them.

Some of the peaks at 4 MHz are aliasing from the frequency limit of our detector, and are a measurement artifact. Others (around 2083 kHz) are from interactions with another mode (as in the supplementary, Sec. F). The aliasing peaks could be removed by improving the detector bandwidth, while the spurious peaks that come from interactions with another mechanical mode could be removed by repositioning the laser.

(2) Fig1 d shows quite a lot of spurious modes, of which the peak heights are comparable and even higher than frequency combs (after the 7th sidebands). This will affect the real case application.

Some of the spurious modes of Fig. 2d (marked by blue arrows in the figure below) are likely related to the near-degeneracy of the 2nd membrane mode (by symmetry). By comparing to 2a and 2e, one can see that modes without this (near-)degeneracy have a cleaner spectrum. The near-degeneracy of these modes would indeed limit applications, but can be readily engineered by making the membrane rectangular instead of square.

There are other spurious modes (green squares), which occur around the 2nd mode \pm fundamental frequency. We suspect these are due to interactions between the fundamental mode and the 2nd mode. At higher frequencies, some peaks are the result of aliasing due to our detection bandwidth.

The benefit of our overtone combs is that the mechanical mode can be selected by positioning the laser. However, if the positioning is not optimal, spurious modes can occur.

(3) I see this work use a trampoline structure SiN resonator. What is the advantage compared to the case that directly uses a whole membrane? Presenting a table of the parameters of the device (e.g., mechanical mode frequency, linewidth, Q value) may improve the readability of the article.

The trampoline membrane is much more compliant than a whole (square) membrane, which allows us to reach larger displacements (i.e. cross more nodes of our optical trap). We have made revisions to the text to make it more readable, but would like to point out that the data in the manuscript comes from different membranes. The relevant parameters (frequency, decay rate) are slightly different between the membranes, so we report the typical design(ed) values in the supplementary, Sec. B.

(4) Can the authors clarify whether the trapped optical field will introduce interactions between different sets of the frequency combs, e.g., interactions between the mechanical frequency combs of the second mode (257 kHz) and third modes (359 kHz).

So far, we have only seen interactions between the comb and a single mode (SI, Sec. F). These can potentially be used to extend the bandwidth of the comb. However, we have not seen interactions between different combs. We have started to study the dynamics of growth/decay of the comb by repositioning the laser spot, but feel that those results are better suited for a separate publication.

Reviewer 2:

1. They use the wording "frequency comb" to describe the observed multiple and equally-spaced spectrum but I am not sure if this terminology is adequate. It is, of course, a matter of definition but majority of frequency comb experiments consist of a high frequency carrier with multiple sidebands with low-frequency interval, which is highly different from the carrier frequency. In contrast, the generated multiple peaks in this work are all high harmonic overtones and I wonder if we can refer them as a frequency comb. It is rather natural to use the words something like "ultrahigh overtone generation".

Indeed, the common use of the terminology 'frequency comb' refers to a carrier with many sidebands, where the phase coherence of the sidebands leads to a pulse train. In our work, the comb teeth are integer multiples of a particular mechanical mode, which is in principle not different from the mode families used in optical combs (Fabry-Perot or whispering gallery modes), nor from the modes of a bulk acoustic wave resonator. The distinguishing feature between any Fabry-Perot cavity (which has equal mode spacing) and a frequency comb is the phase coherence of the different modes. We show in Fig. 4 and supplementary section J that the overtones of our mechanical resonator have a fixed phase relation on top of their fixed frequency relation, which motivates the use of the term 'frequency comb' rather than merely 'overtone generation'. We have made changes to the wording throughout the manuscript to make this distinction more clear.

2. Relating to this first point, the application of this system is unclear. In the introduction part, the authors list up the applications of optical frequency comb, but these applications utilize equally spaced sidebands, not the high harmonic overtones of carrier signals. One of the significant features of frequency comb is the generation of pulse train (or "solitons"), but I believe that the overtones only modify the shape of the waveform from sinusoidal to distorted one, as shown in Fig.4c and d, and do not generate a longer-period pulse train. It is also the case in spectroscopy usages (Ref. [11] and [12]), where I cannot imagine the mechanical counterpart; how to utilize the overtones in the "single" mechanical resonance in, for example, molecular sensing etc. ? The use of this overtone generation should be something different from that of optical frequency comb. I recommend the

authors to modify the introduction and the conclusion parts to justify the importance of "overtone generation".

We agree with the reviewer that the introduction focused too much on the applications of optical frequency combs, which are different from those of mechanical frequency combs. Spectroscopy with mechanical frequency combs is indeed difficult to imagine, but position sensing in optically opaque materials (e.g. underwater, biological tissue, solid materials) would be a promising application in several fields. We have re-written the introduction to more clearly focus on applications of mechanical frequency combs.

For optical combs, the existence of a pulse train demonstrates that the different comb lines have a fixed phase relation, which is necessary for most frequency comb applications. In our work, we show that such a fixed phase relation also exists between the overtones (fig. 4 and fig S18), but due to their frequency relation it does not result in a pulse train.

3. The role of stability of high overtones should be discussed. The stability of frequency comb is managed by system non-linearity or feedback mechanics so that it is worth being characterized and analyzed. However, the stability of this overtone "comb" simply reflects that of fundamental frequency generated by the parametric driving. In other words, the overtone generation itself does not contribute to stabilize the frequency accuracy. If the stability of the used laser is low, the harmonics have worse stability but cannot be better than the carrier oscillation. It is unclear for me how to use the same or worse stability of the mechanical overtones in real timing device applications.

The reviewer is exactly right that the stability of the overtones is simply that of the fundamental mechanical mode. The overtones do not enhance the stability of the fundamental mode, but these high Q-factor mechanical modes are inherently stable in frequency. They could be used to phase stabilize different mechanical modes (see e.g. Ganesan & Seshia, *Scientific Reports* 9 9452 (2019)) with lower Q-factors/more susceptible to dephasing, or serve as a local (on-chip) phase/frequency reference.

Laser frequency drifts do not impact the frequencies of the overtones at all, as these are strictly integer multiples of the mechanical mode. Changes in laser power do directly affect the amplitude of the overtones (via F_o), but do not affect their frequency. The main source of changes in the comb frequency is due to the temperature shifting the mechanical mode frequency.

4. The authors use a Doppler interferometer for detecting the acoustic vibration. It works when the displacement is larger than the wavelength but it is also true that apparent high harmonic overtones are introduced in the output signal with large displacement. Used electronics also can sustain the overtones and the author should experimentally prove the contributions are enough small.

This is a good point and we have now included a thorough verification in the SI (Sec. B) of sources of higher harmonics in a laser Doppler vibrometer. We have verified that it was operated within the specifications, which should guarantee that the measured frequency comb corresponds to the real motion of the resonator. The SI Sec. B has been expanded significantly with references to the literature describing effects in Doppler vibrometers that could lead to spurious harmonics (G. Siegmund, 8th international conference on vibration measurements by laser techniques). Both optical and electronic sources of higher harmonics are described and we provide several new measurements (Fig. S5) and highlight previously included measurements (Figs. S14-S16) to demonstrate that the frequency comb is not an artifact of the setup.

More details to be answered:

5. It is interesting to see the lateral mode shape of the high overtones. Because it is not the eigenfrequency of the resonator, the lateral shape might be different from that of fundamental frequency (see for example ref. [48]). This can be a new aspect and I recommend the authors to discuss it as well as the validity of eq (1).

We agree that the mode shape of the high overtones would be interesting, but we have not succeeded in capturing sufficiently clear mode shapes. We expect the mode shapes of the overtones to be similar to the concentric rings reported in F. Yang et al., PRL **122** 154301 (2019). Although they propose a different explanation in their work, their setup could have the same overtone mechanism as in our work.

In the figure below, we compare the velocity in the center pad of the membrane to the center part of the square membrane from F. Yang et al (middle row). While there are similarities in the shapes, our measurements are hindered by the fact that our laser-beam position greatly affects the frequency comb (even when the thermal parametric driving is absent and we drive the comb with a sine wave). We have made these measurements specifically to reply to this interesting suggestion but do not think these are yet of the quality suited for the paper.

Additionally, we have revised Eq 1 to have the proper form of the gradient force, and removed the drive force for simplicity since it is not necessary for the overtones. The supplementary has been changed to reflect these changes.

6. On the piezo drive shown in the S.I., the white noise excitation might directly excite the overtone oscillation. It is necessary to experimentally confirm that similar intensities of overtones can be generated when only the fundamental mode is piezo-excited.

We have performed additional measurements based on this suggestion, and indeed the overtones can be generated when only the fundamental mode is driven with a sine wave. These results are shown in Fig. S11, and SI Sec. E has been updated accordingly.

7. It is also interesting if the optical parametric drive excites only the fundamental frequency. If I correctly understand the drive mechanism, the parametric drive is not the blue side pumping but the laser light creating the thermal stress with a kind of positive feedback to induce a degenerate parametric drive. Therefore, there is no externally given frequency and any higher mode can be excited.

Indeed, it is due to the laser light modulating the stress (at 2x the dominant frequency of motion). The maximum frequency of the modes that can be excited is (likely) determined by the thermal time-scale of the system (i.e. the time delay between the illumination and temperature curves in Fig. S8). The absence of any externally given frequency/drive tone is one of the main benefits of overtone combs, and allows considerably simpler setups to be used.

8. In the spectrum Fig.2 a, the amplitude of fundamental frequency is -10dB lower than the overtones. I do not understand how this situation is created. Does it relate to the points mentioned above?

There is a significant displacement at DC (due to radiation pressure) which may account for the fundamental mode appearing weaker than it is. Additionally, the shape of the mode might account for some variations in detection efficiency. The fundamental mode is weak in some measurements (Figs. 2a, e), but is clearly the strongest mode in others (Figs. 2c,d, Fig. 3e). In simulations, the fundamental mode is always stronger than the overtones, which points to a measurement artifact. The dynamics of the individual overtones, their growth, decay and relative amplitudes are interesting and we are continuing with measurements in this direction. Given the scope of those experiments, they would fit better in a separate publication.

In summary, although they performed nice experiments and analysis, it is quite unclear for me the importance of this work in real applications. The comparison with optical frequency comb is not suitable to my opinion because the intrinsic properties are different, so that the importance of high overtone generation should be justified. With the present form of manuscript, the importance seems to be not in its application as frequency comb but in the new aspects as nonlinear systems, where very high overtones might create new dynamics of mechanical oscillation but the analysis is not sufficient to justify the publication.

We thank the reviewer for their comments on the experiments and analysis. The introduction and conclusion have been adapted to focus more on mechanical frequency combs and their applications. We have particularly highlighted the benefits that the overtone mechanism offers over other ways of creating mechanical frequency combs (no external frequency reference/drive). The analysis of these overtones within the framework of nonlinear mechanics (i.e. the growth, decay and individual overtone amplitudes) is not the main focus of this work; we hope that the changes to the manuscript make it easier to follow that the main focus is the (novel) mechanism for generating overtones (and its benefits).

REVIEWER COMMENTS

Reviewer #1 (Remarks to the Author):

I noticed that in the revised manuscript, the introduction and conclusion have been rewritten to focus on the applications of mechanical frequency combs and highlight the novelty of the mechanism. Using a weak optical trap, the dielectrophoretic force modulates the membrane's mechanical potential and thus creates multiple overtones integers. I agree that this mechanism has not been revealed before and the method may have advantages, e.g., without requiring feedback control, external drives, and frequency references. The same optical trap could be used as a source to bring the membrane to self-oscillation based on the thermal effect. Optothermal driving has been previously reported [e.g., 47, Nano Lett. 2012, 12, 9, 4681–4686] to introduce the mechanical self-oscillation regime. This method is traditional and is not so striking. But I agree that this technology is initially introduced by the authors to drive the silicon membrane motion frequency combs. Combining these two effects, using an optical trap to create mechanical overtone frequency combs may have advantages, e.g., making them easier to operate or miniaturize. So, the main novelty comes from the way of modulating the mechanical potential and the mechanism to generate the mechanical overtone.

Some small issues:

1. The authors claim that “via a novel mechanism that enables frequency combs without requiring feedback control, external drives and frequency references, or precision optics.” This may allow the authors to give the conclusion that “This makes it more versatile and easier to use than other ways of generating these combs.” As reviewed by the authors [9-10], optomechanical devices could also generate mechanical frequency combs. For an optomechanical generation of the frequency comb, a single laser tone is used to drive and detect, without requiring feedback control, external drives, and frequency references. The operations look also quite easy and the device is on a micro-scale. Optomechanical frequencies have also been reported in reference [Phys. Rev. Lett. 127, 134301 (2021)], [Nanophotonics 2020; 9(11): 3535–3544] as well as using SiN membrane [Phys. Rev. Lett. 128, 153901 (2022)]. Comparisons with these works should be taken into account to reach pertinent conclusions.

2. Regarding the applications of mechanical frequency combs, the authors foresee a vital role for mechanical frequency combs in transduction [32] or coupling to multiple qubits [33]. I am wondering about the mechanical thermal noise. Optothermal driving is useful for mechanical self-oscillation but may be fatal to quantum applications. How to cool low-frequency mechanical modes with modulated potential is not clear.

In a conclusion, the manuscript's presentation is improved. The authors also answered most of the technical issues satisfactorily. The manuscript now looks well organized and contains essential

information on experimental, numerical, and theoretical details. After considering the above issues, I think it is acceptable to the publication.

Reviewer #2 (Remarks to the Author):

I thank the authors to carefully address the issues I pointed, especially to the additional experiments. I believe that the revised manuscript shows enough novelty and importance for publication.

We thank the reviewers for their helpful comments. We have added a reply to the comments (copied below) in blue, and marked changes in our work in the same color. The content of this manuscript is to be included in the PhD thesis of one of the first author. In the preparation to defend the thesis, minors comments were made by one of the committee members (mainly in the supplementary information), which have also been addressed in this version of the manuscript.

Reviewer #1:

Some small issues:

1. The authors claim that “via a novel mechanism that enables frequency combs without requiring feedback control, external drives and frequency references, or precision optics.” This may allow the authors to give the conclusion that “This makes it more versatile and easier to use than other ways of generating these combs.” As reviewed by the authors [9-10], optomechanical devices could also generate mechanical frequency combs. For an optomechanical generation of the frequency comb, a single laser tone is used to drive and detect, without requiring feedback control, external drives, and frequency references. The operations look also quite easy and the device is on a micro-scale. Optomechanical frequencies have also been reported in reference [Phys. Rev. Lett. 127, 134301 (2021)], [Nanophotonics 2020; 9(11): 3535–3544] as well as using SiN membrane [Phys. Rev. Lett. 128, 153901 (2022)]. Comparisons with these works should be taken into account to reach pertinent conclusions.

The optomechanical combs are photonic in nature (i.e. the different comb lines are only present optically), and typically only a single mechanical frequency is present. Our work explicitly uses *mechanical* frequency combs, consisting of multiple mechanical frequencies. While our detection method is optical (laser Doppler vibrometry), we take great care to ensure the detected signal contains only information on the mechanical velocity. This is why we limit ourselves to comparison with other purely mechanical combs in Table S1.

We have made changes to the introduction to make the distinction between optomechanical and purely mechanical frequency combs more clear.

2. Regarding the applications of mechanical frequency combs, the authors foresee a vital role for mechanical frequency combs in transduction [32] or coupling to multiple qubits [33]. I am wondering about the mechanical thermal noise. Optothermal driving is useful for mechanical self-oscillation but may be fatal to quantum applications. How to cool low-frequency mechanical modes with modulated potential is not clear.

The reviewer correctly points out that many applications in the quantum regime require low occupation numbers of the mechanical state, and we are far from the low occupation number (quantum) regime. The individual overtones are not separate modes in our model, where it is not exactly clear how to define an occupation number for each of them. The fundamental mode is strongly driven, so the comb as a whole is far from the quantum regime.

We have shown that the frequency comb interacts with different modes of the same structure (Supplementary Sec F and Fig. S15), and that it can be detected outside the membrane itself (Fig. S7). Combining these two effects could be an avenue to transferring the comb to a different mechanical mode, which could then be cooled. We have rewritten the relevant part of the introduction to better represent the possibilities of mechanical combs for transduction and coupling to multiple qubits.

In a conclusion, the manuscript's presentation is improved. The authors also answered most of the technical issues satisfactorily. The manuscript now looks well organized and contains essential information on experimental, numerical, and theoretical details. After considering the above issues, I think it is acceptable to the publication.

We thank the reviewer for their comments on the comparison with other frequency combs, and on the utility of mechanical frequency combs for quantum applications.

Reviewer #2:

I thank the authors to carefully address the issues I pointed, especially to the additional experiments. I believe that the revised manuscript shows enough novelty and importance for publication.

We would like to thank the reviewer for their kind comment.